# An Empirical Investigation of University Students’ Behavioural Intention to Adopt Online Learning: Evidence from China

**DOI:** 10.3390/bs12100403

**Published:** 2022-10-20

**Authors:** Lu Hai, Guoyuan Sang, Hui Wang, Wenyu Li, Xiaohong Bao

**Affiliations:** 1School of Education, Minzu University of China, Beijing 100081, China; 2Faculty of Education, Beijing Normal University, Beijing 100875, China; 3College of Educational Science, Xinjiang Normal University, Wulumuqi 830054, China

**Keywords:** behavioural intention, perceived usefulness, perceived ease of use, self-regulated online learning, online learning self-efficacy

## Abstract

The present study examined the relationship among behavioural intention (BI) to adopt online learning, perceived usefulness (PU), perceived ease of use (PEU), self-regulated online learning (SR) and online learning self-efficacy (SE). A total of 900 university students with online learning experience from many provinces of China took part in the study. Structural equation modelling (SEM) was used to analyse the data accepted. The results indicate that PU has a significant positive effect on BI; SR has a significant positive effect on PEU, PU and BI. SE has a significant positive effect on PEU, PU and BI. In addition, SE and SR have significant indirect effects on BI through the mediation of PU. The outcomes have tangible theoretical and practical implications. They not only replicates previous research and provides possible space for further expansion of TAM, but also provide us with an opportunity to reflect on and actively take practical measures to improve BI. These efforts include teachers, parents and other educators trying to promote students’ academic achievements, self-efficacy and self-regulation in the process of online learning. The former is the most concerning issue, while the latter two are the source of students’ motivation. Furthermore, educators should make appropriate use of the role of digital technology in online learning and be careful not to exaggerate the value of digital technology, let alone equate it with online learning.

## 1. Introduction

With the rapid development of internet technologies, online learning is becoming increasingly popular with higher education. Furthermore, the outbreak and widespread coronavirus disease 2019 (COVID-19) also influenced the education system [1], and a growing number of universities around the globe have rapidly transitioned from offline to online courses [1,2]. According to survey data, 1454 colleges and universities offered 12.26 million online courses, and approximately 1 million teachers and over 17 million students participated in online learning between February and June 2020 in China [3]. At the same time, as some scholars have pointed out, the growth of online learning will continue [4].

As opposed to traditional face-to-face learning, online learning refers to learning that takes place partially or entirely on the internet [5] and has several forms. The online learning in this article refers to classroom learning with the help of technology. Generally, students in online learning conditions are viewed as performing better than those receiving offline learning [5,6], which is partly due to the advantages of online learning; for example, online learning can be better adapted to students’ diverse needs by breaking down geographical and physical barriers, and learners can adjust the pace of instruction, moving quickly through familiar material and slowing down when needed [7,8]. Online learning offers rich educational resources, presented in the form of multiple media [5]. Asynchronous communication makes students to be more thoughtful before answering questions, and online materials often ask participants to ‘share their thoughts’ in their postings, which is highly conducive to deep learning [9]. Students feel comfortable with social media and online learning methods for academic study and courses generally [10]. The factors mentioned above can to some extent predict the behavioural intention to adopt online learning [11]. Undoubtedly, most of these studies are limited by the perspective of nonlearners, such as a technology-centred perspective, and pay little attention to the initiative and responsibility of the students who are the protagonists of the learning process of online learning, although existing studies have pointed out that self-related thoughts and behaviours (e.g., self-efficacy, self-regulation) play an important role in learning processes [12,13] and influence online learning intention [8,14]. To address the research gap stated above, this study analyses the link between some factors related to online learning. These include a selection of factors from the technology acceptance model (TAM) [15] focusing on the technical perspective, two additional factors (self-efficacy and self-regulation) focusing on the students’ perspective, and by assuming a new relationship among these factors. The result helps to enhance our ability to interpret the factors that may influence behavioural intention, perceived usefulness, and perceived ease of use, eventually improving students’ academic achievement and willingness to use online learning by taking effective measures.

## 2. Theoretical Framework and Literature Review

### 2.1. Technology Acceptance Model as Theoretical Framework

TAM, introduced by Davis [16], is an adaptation of the theory of reasoned action specifically tailored for modelling user acceptance of information systems [15]. TAM is shown in Figure 1, with arrows representing causal relationships. According to the model, usage is determined by behavioural intention to use, which in turn is jointly determined by both attitude towards using an object and its perceived usefulness. Attitude towards use is a function of two major beliefs: perceived usefulness and perceived ease of use. Perceived usefulness can be affected by various external variables and perceived ease of use. External variables have a causal effect on perceived ease of use [15,16].

This theory has been used and validated in many studies on technology adoption and acceptance (e.g., [17,18]), and it is even regarded as the most widely used and influential framework for exploring people’s attitudes and intentions to adopt technology [2,19]. In addition, TAM has also been an appropriate model as a theoretical framework in research settings in an educational context [20]. Therefore, this paper adopts the TAM as its theoretical framework. To make it more relevant to this study, we modified it by selecting behavioural intention, perceived usefulness and perceived ease of use from the model and adding self-regulation and self-efficacy from the students’ perspectives.

### 2.2. Behavioural Intention to Adopt Online Learning (BI)

Behavioural intention is viewed as a proxy to predict actual usage [21]; more specifically, if a person intends to perform a behaviour, then it is likely to be done [22]. As a result, understanding students’ behavioural intention to adopt online learning contributes to understanding their actual behaviour. In addition, as mentioned above, behavioural intention has been taken as a dependent variable in many relevant studies (e.g., [8,23]). Therefore, it is also used as a dependent variable in this paper, but we made a modification to fit the theme and expressed it as a whole as by “behavioural intention to adopt online learning”.

### 2.3. Perceived Usefulness(PU)

Perceived usefulness is defined as “the degree to which an individual believes that using a particular system would enhance his or her job performance” [16], which is similar to performance expectancy [24]. Perceived usefulness refers to the individual’s perceived effect of online learning on academic performance in the current research. According to prior research, perceived usefulness has a positive impact on adoption intention [25,26] and is a statistically significant predictor of behavioural intention [27]. It may also mean that higher perceived usefulness results in higher behavioural intention to adopt online learning. Based on this literature, we establish the first hypothesis.

**H1:** *PU will have a positive effect on BI*.

### 2.4. Perceived Ease of Use(PEU)

Perceived ease of use is defined as “the degree to which an individual believes that using a particular system would be free of physical and mental effort” [16], which is similar to effort expectancy [24]. This concept refers to the literacy of students to use digital media and technology for successful online learning in the current research. Davis argued that perceived ease of use had a significant direct positive effect on perceived usefulness, as a system that is easier to use will, all else being equal, improve user performance (i.e., greater usefulness) [16]. This is consistent with the research of Dasgupta, Granger and McGarry [17] and [18]. In fact, technical problems have long posed challenges to the use of technology for learning [4], so that, if online learning technology is easy to use, students will realise its additional benefits. Based on this literature, we establish the second hypothesis.

**H2:** *PEU will have a positive effect on PU*.

The results from the extant literature also indicate that perceived ease of use was a statistically significant predictor of intention to use the internet (e.g., [25,27]). From the students’ perspective, they will develop a positive attitude towards online learning and increase their intention to use it if they perceive that the technology involved in online learning is simple. Based on this literature, we establish the third hypothesis.

**H3:** *PEU will have a positive effect on BI*.

### 2.5. Online Learning Self-Regulation (SR)

Self-regulation is defined as a set of principles and practices by which people monitor their own behaviours and consciously adjust those behaviours in pursuit of personal goals [19], including metacognitive strategies, management and control of effort and actual cognitive strategies [28]. Self-regulated online learning is very important to the study of the development of educational systems with computer-assisted technology, especially in online learning contexts [29], and is even viewed as a key element of online learning [30].

Chen and Hwang separated self-regulation into two parts, metacognition and motivation, and found that they had significant positive effects on effort expectancy [29]. This is because online learning self-regulation, as actions and processes directed at acquiring information or skills that involve agency, purpose, and instrumentality perceptions by learners [31], had a profound impact on individuals’ cognition of the difficulty of things. for instance, in a learning environment with a considerable degree of autonomy, where students with lower levels of self-regulation experience greater difficulties [32]. That is, the higher students’ self-regulation ability is, the higher their perceived ease of using technology in online learning. Based on this literature, we establish the fourth hypothesis.

**H4:** *SR will have a positive effect on PEU*.

Students’ self-regulation is crucial to academic achievement in the course [33], possibly because students who have self-direction and self-regulation ability can manage their own learning processes and have more positive beliefs about the effectiveness of online learning [30,34]. This enables students to adopt correct learning strategies, constantly reflecting and improving learning efficiency, and eventually obtaining better academic performance. Based on this literature, we establish the fifth hypothesis.

**H5:** *SR will have a positive effect on PU*.

Self-regulated learning capability has been shown to shape individuals’ attitudes towards online learning and has a significant positive effect on participants’ continuous intention to learn online [8]. This finding is supported by Hood, who concluded that higher self-regulation of time and the study environment were closely associated with greater intention to use archived virtual tutorials, and that higher self-regulation was the most important predictor of intentions to attend online lectures [14]. Based on this literature, we establish the sixth hypothesis.

**H6:** *SR will have a positive effect on BI*.

### 2.6. Online Learning Self-Efficacy (SE)

Perceived self-efficacy is concerned with judgements of how well one can execute courses of action required to deal with prospective situations [12], which was regarded as the first most commonly used external factor when Abdullah and Ward analysed 107 papers using external factors of TAM in the context of e-learning adoption [35]. Online learning self-efficacy refers to an individual’s confidence in their ability to successfully embrace online learning and achieve good academic performance.

Self-efficacy was the best predictor of students’ perceived ease of using e-learning systems [35], which is supported by recent studies indicating that perceived self-efficacy had a significant influence on students’ perception of the ease of usage [36,37]. In other words, students with a high level of self-efficacy have a more positive attitude towards themselves, spend more time studying certain internet technology tools and are familiar with their interfaces and features, are more likely to overcome possible technology difficulties and ultimately experience the simplicity of the tool. Based on this literature, we establish the seventh hypothesis.

**H7:** *SE will have a positive effect on PEU*.

Evidence about the relationship between self-efficacy and perceived behavioural usefulness lacks consistency. Abdullah and Ward carried out a systematic analysis of 27 related articles and found that more than half of them lacked a positive significant correlation [35], which is consistent with the findings of Wu, Li, Zheng and Guo [38]. Nevertheless, it was found that online learning self-efficacy had a significant positive impact on the use and user satisfaction of e-learning in some studies (e.g., [39,40]). The latter is more reasonable based on Bandura’s theory [12]; that is, self-efficacy determines how much effort people will expend and how long they will persist in the face of obstacles or aversive experiences, and so favourable outcomes occur if individuals have high self-efficacy. Based on this literature, we establish the eighth hypothesis.

**H8:** *SE will have a positive effect on PU*.

The existing body of literature on online learning reported that self-efficacy was a critical factor in students’ behavioural intentions to use online learning courses on the internet [11]. This was confirmed by recently published studies (e.g., [36]) which showed that self-efficacy positively influences the perceived intention to use online learning systems. Based on this literature, we establish the ninth hypothesis.

**H9:** *SE will have a positive effect on BI*.

In addition, according to the relationship among BI, PU, PEU, SR and SE mentioned in the above literature, we believe that these variables play an indirect role in the relationship between their two adjacent variables. Therefore, we propose the following hypothesis:

**H10:** *PU will mediate the association between PEU and BI*.

**H11:** *PEU will mediate the association between SR and PU*.

**H12:** *PEU will mediate the association between SE and PU*.

**H13(1):** *PU will mediate the association between SR and BI*.

**H13(2):** *PEU will mediate the association between SR and BI*.

**H13(3):** *PEU and PU will mediate the association between SR and BI*.

**H14(1):** *PU will mediate the association between SE and BI*.

**H14(2):** *PEU will mediate the association between SE and BI*.

**H14(3):** *PEU and PU will mediate the association between SE and BI*.

## 3. Method

### 3.1. Participants

A total of 900 students were recruited and completed questionnaires. One hundred and thirty-three (14.78%) questionnaires were excluded from further analysis due to poor response quality. The final sample included 767 participants, and their responses were used for data analysis, including confirmatory factor analysis, descriptive analysis, correlation analysis and structural equation modelling (SEM). Detailed demographic information of the participants is shown in Table 1. All participants had undertaken the online learning courses provided by the university and were mainly from the Chongqing, Shandong, Xinjiang, Guangxi, Qinghai and Xizang provinces of China. Data were collected through Wenjuanxing (https://www.wjx.cn/, accessed on 10 September 2021; a popular online survey platform in China) between 10 September and 20 October 2021. The information sheet and the link to the survey were sent to potential participants via WeChat, a widely used instant messaging and social media application in China. We had to give out online questionnaires because offline face-to-face investigation was not allowed due to COVID-19. Participants indicated their consent by completing the questionnaire. It took up to 10 min to complete the survey.

### 3.2. Instrument

The instrument used in the current study consisted of two parts. Part one was named demographical information, which focuses on collecting basic information about the participants, including gender, nationality and grade. Part two was named the online learning scale, which consisted of five subscales covering BI, PU, PEU, SR and SE.

The BI subscale was designed based on Davis [15] and Song, Huang and Li [41], which takes into consideration behavioural intention towards online learning. Six items are included, e.g., I think online learning is a very good experience. The PU subscale was designed based on Zhang, Huang and Li [42] and Tian, Feng and Han [43] to examine the learning effect achieved through online learning and includes 5 items, e.g., I can accurately state the content of this online teaching. The PEU subscale was compiled by the author to examine students’ ability to use digital resources in the process of online learning, which includes 6 items, e.g., when I have a problem with digital technology, I can solve it quickly. Self-regulated online learning and online learning self-efficacy were designed based on Pintrich and De Groot [28] to understand students’ self-regulation ability and perceived confidence during online learning, which includes 3 items and 5 items respectively, e.g., I often push myself to study online and I believe I have the ability to carry out online learning. All items were measured using a five-point Likert scale, ranging from 1 (strongly disagree) to 5 (strongly agree).

### 3.3. Data Analysis

Data were analysed with SPSS 21.0 and AMOS 24.0. First, confirmatory factor analysis (CFA), reliability and validity were calculated to assess the degree of fit, stability and effectiveness of the subscales. Second, descriptive analysis and correlation analysis was calculated to investigate the basic level of the subjects. Third, SEM was conducted to investigate the relationships among the five variables.

## 4. Results

### 4.1. Assessment of the Model Fit

The fit of the research models was assessed, and the results showed that χ^2^/df (chi-square divided by the value of degree of freedom) = 4.07, CFI (comparative fit index) = 0.91, TLI (Tucker–Lewis index) = 0.90, SRMR (standard root-mean-square residual) = 0.05, RMSEA (root mean square error of approximation) = 0.06, indicating a good fitness of the collected data with the measurement model based on the principles of Bentler and Bonett [44], Hu and Bentler [45], MacCallum, Browne and Sugawara [46].

### 4.2. Analysis of Reliability and Validity

The analysis of reliability and validity was conducted for each subscale. Internal consistency reliability was used to examine the correlations among items of the same subscale. The results showed that the internal consistency coefficients of the five subscales were 0.88, 0.76, 0.88, 0.73 and 0.84, which were above the minimum acceptable level of 0.70 [47], denoting high internal consistency for all subscales. Discriminant validity was used to test the extent to which a subscale differs from other subscales, with the square root of the number AVE on the diagonal of the scale mostly higher than the Pearson correlation among subscales, indicating that five subscales have high discriminant validity based on the principle of Fornell and Larcker [48]. See Table 2 for details.

### 4.3. Descriptive Statistics and Correlations among the Variables

To check the quality of the data, skewness and kurtosis for all the measured variables were calculated. The skewness and kurtosis of BI, PU, PEU, SR and SE are −0.33, −0.26, −0.03, −0.32, 0.04; 0.42, 2.46, 0.51, 0.61, 0.55, respectively, denoting that the shape of the data distribution in the study may not be severely nonnormal because of the absolute values of skewness ≤ 3.0 and kurtosis ≤ 10.0 [47].

The means of BI, PU, PEU, SR and SE ranged from 3.42 to 3.51, and the standard deviations ranged from 0.53 to 0.76. Meanwhile, correlations were examined to check whether there was a correlation among these subscales, and it was found that all subscales were significantly (at the alpha level of 0.01) correlated with the range of 0.35–0.56. See Table 3 for details. Since SE and SR are highly correlated, we checked for multicollinearity issues for their parameters, and found that the VIF values were 1.39. As the VIF values were far below the threshold of 10, collinearity was not a concern in the model [47]. At the same time, we also conducted common method bias testing; specifically, we compared the goodness-of-fit indices across three models. The single-factor model used all items of SE and SR as indicators of one latent common method factor; the two-factor model treated SE and SR as two latent factors indicated by their corresponding items; the three-factor model added a second-order common method factor to the two-factor model. The results of single-factor test indicated that the fit of the single-factor model was unsatisfactory: χ^2^/df = 12.24, CFI = 0.90, TLI = 0.85, RMSEA = 0.12. The fit of the two-factor model was acceptable, χ^2^/df = 2.75, CFI = 0.99, TLI = 0.98, RMSEA = 0.05. The model fit of the three-factor model was also acceptable: χ^2^/df = 1.29, CFI = 1.00, TIL = 1.00, RMSEA = 0.02. These results indicated that CMV is not significant and does not account significantly for the shared variance between SE and SR in our data because the single-factor model did not show a good fit to the data and was worse than the two-factor model; meanwhile, latter does not show obviously more deterioration than the three-factor model (Δχ^2^/Δdf = 4.77, *p* > 0.05) [49]. 

### 4.4. Hypothesis Testing

Gender, nationality, and grade were statistically controlled in the tested models as there were significant differences between gender, nationality and grade in the mean scores of some variables, as seen in Table 4. For example, male and Han participants had significantly higher PEU than female and minority participants (*p* < 0.01), respectively. PU scores were significantly greater in the seniors than in the freshmen (*p* < 0.001), sophomores (*p* < 0.01) and juniors (*p* < 0.001); PEU scores were significantly greater in the seniors than in the freshmen (*p* < 0.001), sophomores (*p* < 0.01) and juniors (*p* < 0.01); PEU scores were significantly greater in the sophomores than in the freshmen (*p* < 0.05); SR scores were significantly greater in freshman than in sophomore (*p* < 0.05) and juniors (*p* < 0.01); SE scores were significantly greater in the seniors than in the freshmen (*p* < 0.05).

The significant correlations among BI, PU, PEU, SR and SE showed that these factors were highly connected. To obtain the specific causality relationships among factors, covariance-based SEM was selected based on such specific conditions of choice of the SEM technique, such as the focus, sample size, data normality, etc [50,51]. Additionally, the nonstandardised coefficient (B), standard error (SE), standardised coefficient (β), critical ratio and significance level were calculated to test the abovementioned hypotheses with gender, nationality and grade as control variables. The results indicated that some hypotheses were supported; see Table 5 and Table 6 for details.

The data in Table 5 show that PU has a significant positive direct effect on BI (β = 0.349, *p* < 0.001); therefore, Hypothesis 1 is supported. SR has significant positive direct effects on PEU (β = 0.171, *p* < 0.001), PU (β = 0.377, *p* < 0.001) and BI (β = 0.135, *p* < 0.01); thus, Hypothesis 4, Hypothesis 5 and Hypothesis 6 are supported. SE has significant positive direct effects on PEU (β = 0.507, *p* < 0.001), PU (β = 0.350, *p* < 0.001) and BI (β = 0.411, *p* < 0.001), meaning that Hypothesis 7, Hypothesis 8 and Hypothesis 9 are supported. However, PEU was not found to have significant direct effects on PU (β = 0.048, *p* > 0.05) and BI (β = 0.024, *p* > 0.05); therefore, Hypothesis 2 and Hypothesis 3 were rejected.

To further clarify the mechanism of causality between the nonadjacent subscales, several main paths were analysed. The statistical significance was tested by setting the bootstrap number to 5000 and the confidence interval to 95%. The results revealed that SR had a statistically significant indirect effect on BI through the mediation of PU (β = 0.131, *p* < 0.01). SE had a statistically significant indirect effect on BI via the mediation of PU (β = 0.122, *p* < 0.01), supporting H13(1) and H14(1). Other hypotheses on indirect effects were rejected. See Table 6 for details.

## 5. Discussion

The findings demonstrate that PU has a significant positive effect on students’ BI to adopt online learning, which coincides with the results of Fusilier and Durlabhji [27] and Hong, Thong, Wong and Tam [52], revealing the strong effects of PU on BI. That is, students are more likely to adopt online learning when they achieve good academic performance during online learning. In addition, the results of the present study demonstrate that PU is not only a direct predictor of BI but also a crucial mediator of BI. More specifically, PU played a significant mediating role in the relationship among SE, SR and BI, and its proportion to the total effect was larger. Given the importance of PU, schools should pay more attention to college students’ learning effects and take more measures to promote their academic achievement in the process of online learning, which can effectively improve students’ behavioural intentions to adopt online learning.

The unanticipated finding of this study is that PEU does not have a significant positive effect on PU and BI. This is not consistent with most studies, which show PEU as having a significant effect on students’ PU and BI when using the e-learning system (e.g., [20,53]), but it does not mean it is an entirely divergent result because a handful of studies have also found it (e.g., [36]). This finding means that a high PEU does not directly convert to a high PU and BI. One possible explanation for this result is that PEU in the article only focuses on the technology of online learning process, while PU and BI not only focus on online learning technology, but also on the benefits and good experiences brought by online learning and which are covered in PEU. Thus, improving PEU itself is not enough to improve students’ PU and BI. The result also provides some illuminating reflections; for instance, teachers and parents should avoid a one-sided view of technology orientation and attach importance to knowledge acquisition and competence development, while helping students become more proficient in the use of computer technology.

Similar to the results of previous studies (e. g., [54,55]), our results support the effect of SR on PEU and PU and BI, indicating that self-regulated online learning plays a crucial role in students’ perceived ease of using technology, academic achievement in the online learning process and behavioural intention to adopt online learning. This result can be attributed to the value of self-regulation itself; i.e., when they have high SR online learning, individuals can actively overcome technical trouble and academic problems and achieve academic performance by spending time adjusting learning strategies, which further improves students’ behavioural intention to adopt online learning. Therefore, specific measures such as promotion of students’ perceived leadership and group cohesion [13] should be considered to equip university students with stronger self-regulatory skills. At the same time, SR also has significant indirect effects on BI via PU, which is helpful to explain the relationship between SR and BI in depth. Specifically, BI is a complex psychological phenomenon that may be affected by many factors, meaning that as an isolated individual ability SR needs a process to affect BI. For example, individuals with self-management ability have better control over time, learning strategies and effort. The former is conducive to individual academic performance, while the latter can also improve individuals’ willingness to adopt online learning, as mentioned above. This study found that SE had a significant effect on PEU, PU and BI, which was in accord with the findings of Ifinedo [37], and Lynch and Dembo [56]. At the same time, SE also had a significant indirect effect on BI through PU. Judgements of self-efficacy determine how much effort people will make and how long they will persist in the face of obstacles or aversive experiences. The higher the level of perceived self-efficacy, the more attention and effort are distributed to solve obstacles and finish tasks, and the greater the performance accomplishments [12]. That is, individuals will spend more time studying related technology and overcoming academic difficulties in the online learning process if they have high online learning self-efficacy. This means that they are more likely to have higher technical competence and academic achievement and more active behavioural intentions to adopt e-learning. In contrast, individuals with low online learning self-efficacy cannot overcome any challenges they may face if the work is complex, and they will not persevere in their efforts when suffering setbacks [53]. They may eventually face problems such as poor employment of technology and poor academic performance. Therefore, it is vital to make students confident in the process of online learning. To this end, teachers and parents should adopt measures to help students achieve high academic achievement and provide verbal persuasion and opportunities to observe the outstanding performances of others [12].

## 6. Limitations and Conclusions

Three limitations in the current research should be noted. First, a convenience sampling method was used to collect data and the number of participants was limited, which resulted in the data in the research not being representative of all Chinese university students. To explore the relationships among variables more deeply, future studies should adopt a more comprehensive sampling method and recruit more subjects from more varied provinces in China. Second, although SE and SR were selected in this study from the perspective of students, BI, PU and PEU are influenced by many factors, including family socioeconomic status, parents’ education notions, family atmosphere and the design of the online curriculum offered by the school. Therefore, future research should cover as many different variables as possible. Finally, the cross-sectional study used in this study cannot infer causality between variables in a strict sense, and longitudinal designs or randomised controlled studies can be used in future relevant studies to further verify the relationship between them.

This study uses TAM as a theoretical model and explores the relationship among BI, PU, PEU, SR and SE based on data from Chinese university students. Results of the current study may make several theoretical and practical contributions. From a theoretical perspective, the result replicates previous research and adds evidence to confirm the relationships among PEU, PU and BI. Furthermore, the current study extends this kind of research by documenting the relationships among SE, SR and PEU, PU, BI, which provides possible space for further expansion of TAM to some extent. From a practical perspective, our findings provide us with an opportunity to reflect on and actively take practical measures to improve PEU, PU and especially BI. For example, teachers, parents and other educators should properly recognise that digital technology plays a supporting rather than central role in the process of online learning, and thus avoid laying much more stress on it than on online learning. In addition, some measures should be taken to promote students’ online learning self-efficacy and self-regulation, such as giving students training of an outstanding quality and a technical support team that includes experts in using the system [53], providing complex, personally meaningful tasks in which students have multiple opportunities for decision making, autonomy, self- and peer evaluation, and can work collaboratively [57]. Perhaps most importantly, teachers should always pay attention to improving students’ academic performance.

## Figures and Tables

**Figure 1 behavsci-12-00403-f001:**
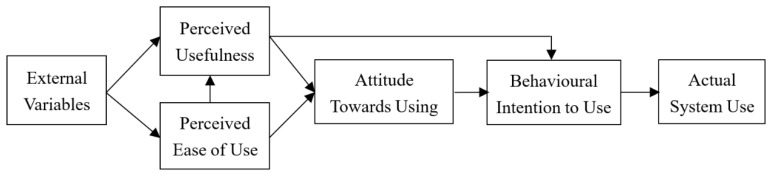
Technology Acceptance Model (TAM).

**Table 1 behavsci-12-00403-t001:** Demographic information of the participants.

Demographic Variable		Sample	
		Number	Percentage
Gender	male	229	29.9
	female	538	70.1
Nationality	Han	457	59.6
	minority	310	40.4
Grade	freshman	370	48.2
	sophomore	156	20.3
	junior	119	15.5
	senior	122	15.9

**Table 2 behavsci-12-00403-t002:** Reliability and validity of all subscales.

	Internal Consistency Coefficient	Discriminant Validity				
		BI	PU	PEU	SR	SE
BI	0.88	0.74				
PU	0.76	0.62	0.63			
PEU	0.88	0.40	0.36	0.74		
SR	0.73	0.56	0.56	0.44	0.69	
SE	0.84	0.65	0.53	0.57	0.67	0.72

**Table 3 behavsci-12-00403-t003:** Descriptive statistics and correlation coefficients among subscales.

Subscales	M	SD	Correlation Coefficient				
			BI	PU	PEU	SR	SE
BI	3.51	0.76	1.00				
PU	3.43	0.53	0.50 **	1.00			
PEU	3.38	0.73	0.39 **	0.35 **	1.00		
SR	3.39	0.72	0.46 **	0.44 **	0.39 **	1.00	
SE	3.42	0.66	0.56 **	0.45 **	0.51 **	0.53 **	1.00

Notes: ** *p* < 0.01.

**Table 4 behavsci-12-00403-t004:** Differences test BI, PU, PEU, SR and SE by gender, nationality and grade.

		BI		PU		PEU		SR		SE	
		M	SD	M	SD	M	SD	M	SD	M	SD
Gender	male	3.49	0.86	3.41	0.65	3.51	0.76	3.43	0.80	3.45	0.72
	female	3.52	0.72	3.44	0.48	3.32	0.71	3.37	0.68	3.41	0.63
	t	−0.54		−0.54		3.35 **		0.93		0.66	
Nationality	Han	3.54	0.80	3.45	0.56	3.44	0.76	3.41	0.73	3.46	0.69
	minority	3.47	0.70	3.40	0.49	3.29	0.67	3.36	0.70	3.37	0.61
		1.23		1.16		2.93 **		1.06		1.91	
Grade	freshman	3.52	0.75	3.41	0.52	3.27	0.71	3.47	0.72	3.38	0.59
	sophomore	3.50	0.79	3.40	0.58	3.42	0.75	3.33	0.75	3.46	0.72
	junior	3.41	0.80	3.34	0.46	3.39	0.70	3.23	0.64	3.38	0.69
	senior	3.61	0.71	3.61	0.56	3.64	0.72	3.38	0.70	3.57	0.70
	F	1.38		6.11 ***		8.37 ***		3.91 **		2.88 *	

Notes: * *p* < 0.05, ** *p* < 0.01. *** *p* < 0.001.

**Table 5 behavsci-12-00403-t005:** The results of structural equation modelling.

Hypothesis	Hypothesised Path	B	SE	β	Critical Ratio	Result
H1	PU→BI	0.714	0.103	0.349	6.940 ***	Supported
H2	PEU→PU	0.021	0.022	0.048	0.978	Rejected
H3	PEU→BI	0.022	0.038	0.024	0.585	Rejected
H4	SR→PEU	0.222	0.062	0.171	3.599 ***	Supported
H5	SR→PU	0.218	0.034	0.377	6.465 ***	Supported
H6	SR→BI	0.160	0.058	0.135	2.751 **	Supported
H7	SE→PEU	0.584	0.055	0.507	10.661 ***	Supported
H8	SE→PU	0.181	0.030	0.350	5.937 ***	Supported
H9	SE→BI	0.433	0.054	0.411	8.019 ***	Supported

Notes: ** *p* < 0.01, *** *p* < 0.001.

**Table 6 behavsci-12-00403-t006:** Total and indirect effects and ratio of both.

Hypothesis	Hypothesised Path	Total Effect	Indirect Effect	Result	Ratio of Indirect Effect to Total Effect
H10	PEU→PU→BI	0.041	0.017	Rejected	0.415
H11	SR→PEU→PU	0.385 *	0.008	Rejected	0.021
H12	SE→PEU→PU	0.374 *	0.024	Rejected	0.064
H13(1)	SR→PU→BI	0.273 **	0.131 **	Supported	0.480
H13(2)	SR→PEU→BI		0.004	Rejected	0.015
H13(3)	SR→PEU→PU→BI		0.003	Rejected	0.011
H14(1)	SE→PU→BI	0.554 *	0.122 **	Supported	0.220
H14(2)	SE→PEU→BI		0.012	Rejected	0.022
H14(3)	SE→PEU→PU→BI		0.008	Rejected	0.014

Notes: * *p* < 0.05, ** *p* < 0.01.

## Data Availability

Data supporting reported results are available from the authors on request.

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
