# Peer review of "An Empirical Investigation of University Students’ Behavioural Intention to Adopt Online Learning: Evidence from China"

_behavsci, 2022, doi:10.3390/bs12100403_

Round 1

Reviewer 1 Report

Dear authors,

The research is interesting and up-to-date. However, I would suggest a clearer research methodology.
How did you calculate the sample?
How many questions does the questionnaire have? What are the types of questions used?Are there any respondent filtering questions?
The limits of the research must be written before the conclusions.

Also, the research conclusions can be broader.

Author Response

Thank you very much for your review and constructive comments. We have made revisions according to your comments. The details are as follows.

RESPONSE to the 1st comment:

In this study, the sample size is 900. Because of the widespread of COVID-19, the selection of samples and the way of carrying out investigation are affected. We had to give out online questionnaire because offline face-to-face investigation is not allowed because of COVID-19, so it is not easy and convenient to get reliable samples in large scale.

Before we begin the investigation, we preliminarily determined the reference value of the sample size according to Gpower and traditional practices (for example, the number of samples should be at least 5 to 10 times the total number of survey scale items).  There are 25 items in our questionnaire, so the sample number should be at least 125 to 250. If we take the total number of college and university students in China into consideration, 125 to 250 participants will not be representative, thus we thought the sample should be larger than that. Then we drew experience from other similar studies. Finally, limited by time and cost, we got 900 questionnaires, among which 767 are valid. We’ve done our best to make samples representative in order to make the study result reflecting the fact.

RESPONSE to the 2nd comment:

(1) The questionnaire includes a total of 25 items, 6 items about behavioral intention to adopt online learning, 5 items about perceived usefulness, 6 items about perceived ease of use, 3 items about self-regulated online learning and 5 items about online learning self-efficacy.

(2) All items were presented to the subjects in the form of self-report.

(3) Due to the small number of items in the questionnaire, we did not set any specific filtering questions. However, we deleted invalid questionnaires by the quality of the questionnaires submitted by the subjects, eg, the subjects responded the same to all items.

RESPONSE to the 3rd comment:

Thank you very much for your suggestion. We adjusted the position of the limitations and conclusion, and we put limitations before the conclusion.

RESPONSE to the 4th comment:

We agree with your opinion. We have revised and improved the conclusion section in line 438-454.

Reviewer 2 Report

This paper shows the results about an empirical investigation of university students’ behavioural intention to adopt online learning and I consider this topic quite useful to fuel the debate among scientific community.

However, may I suggest to the authors:

-       Indicate why aspects such as motivation and collaborative learning were not considered. At p. 2 the authors state that “To make it more relevant to this study, we modified it by selecting behavioral intention, perceived usefulness and perceived ease of use from the model and adding self-regulation and self-efficacy from the students’ perspective”. Why were only these two aspects included?

-       It would be interesting to have a description through which the survey items were attributed to the categories (BI, PU, PEU, SR SE). In the result at page 6 is mention only how many item but not which is included in each category.

The article is well written and a clear research process is outlined.

Author Response

Thank you very much for your review and constructive comments. We have made revisions according to your comments. The details are as follows.

RESPONSE to the 1st comment:

As an important topic, behavioral intention to adopt online learning is related to many variables, including motivation and collaborative learning you have mentioned. However, due to the requirement of simplicity of the article and to relieve the cognitive pressure of the subjects when completing the questionnaire, we can only purposefully select one or two variables. This paper chooses self-efficacy and self-regulation because they are very classic concepts in psychology, and they are closely related to the learners themselves that this paper focuses on. Of course, we will pay attention to more variables in future studies, including the two variables you mentioned, to better explain the relationship between different relevant variables and, to provide effective suggestions of promoting intention to adopt online learning.

RESPONSE to the 2nd comment:

We have reported some specific items included in each dimension in line 258-268 by way of providing examples.

Reviewer 3 Report

The topic is an exciting and timely need. This manuscript's overall flow suits the journal's focus well. However, some specific aspects of this manuscript need to be considered before the decision for publication. See below, 

1. In line 25, the number of online courses and students who took the courses are similar. Would you recheck the numbers? 

2. In lines 97 and 249, the spacing does not look correct. 

3. In lines 192 and 228, you ran the EFA to choose the items for each factor with specific criteria. However, the purpose of EFA is not to select items. It determines the factors from the dataset when you do not have any structure under the data. Even though you have the string theoretical model, why would you decide to run the EFA? If you have any reference to show the justification, it should be added to the text. If not, you may revise the data analysis method.  

4. In lines 192 and 194, you mentioned sample 1 and sample 2. Why did you separate the samples? Provide the reason for that. 

5. In line 186, this hypothesized model is derived from the latent factors' structure. Therefore, it would be better to use circles than rectangles. 

6. In line 204, the item numbers for each factor can be mentioned in this section. 

It would be improved if you clarified those considerations.  

Author Response

Thank you very much for your review and constructive comments. We have made revisions according to your comments. The details are as follows.

RESPONSE to the 1st comment:

We have rechecked the numbers and provided more specific details (line 27-29).

RESPONSE to the 2nd comment:

We have checked and revised the spacing in lines 97 and 249 (line in 113 and in 300 in the revised version). We also checked and revised the problem of wrong spacing in other parts of the article.

RESPONSE to the 3rd comment:

After reading the relevant literature (such as Fabrigar, Wegener, MacCllum, & Strahan, 1999), we found that your opinion is reasonable. We already had a clear sub-questionnaire when we distributed the questionnaire, so we should directly run the CFA rather than the EFA. Thus, we have removed the EFA section and its relevant contents.

RESPONSE to the 4th comment:

As mentioned above, we mistakenly ran EFA, and sample 1 was mainly used for EFA, which is why we divided the subjects into sample 1 and sample 2 before. We have removed the division of subjects and only used sample 2 after we deleted EFA.

RESPONSE to the 5th comment:

We agree with you that the variables in the model are latent variables rather than observed variables. So we decided to represent them as circles. However, we did not make changes to Figure 1, because it was presented by rectangle form in original document.

RESPONSE to the 6th comment:

Thank you very much for your suggestion. We added the item numbers for each factor in line 258-268.

In addition, the manuscript has been checked by a colleague fluent in English writing.

Reviewer 4 Report

It is a well-crafted study with clear presentations of the facts and a detailed discussion of the results and analysis. Enjoyed reading through the manuscript in detail. Authors have put-in good efforts. They are advised to improve the study in the light of below suggestions.

1- Online learning modes often report such issues as loss of interest, performance degrade and cyberslacking for the students as opposed to normal classroom learning. Authors should make a distinction here whether they have tried to survey students who studied online via MOOC or online websites or it was classroom learning through the use of technology.

2- The introduction section lacks in clear presentation about the existing research gap, novelty and contributions of the research. I believe the introduction section needs revision accordingly. Similarly the basis for introduction of self-efficacy and self-regulation is week and need to reviewed here.

3- Covariance-based SEM method was used for the analysis. What is the rationale for that? What is the criterion for sample size and sample for the two study samples? Students from different provinces filled the data, so that might be a concern in terms of bias. Under such circumstances, it is advisable to fix quota for different regions. 

4- No mention of control variables is discussed. Were they tested or controlled for in the study? The students experience and ease w.r.t online learning are another point of concern here. A sophomore may not be well-experienced with online learning as compared to a senior student so this should be controlled for in the study. The level of seniority here might be influencing the SE and SR factors as well.

5- All the hypotheses have been put together at the end of literature part. It is advisable to put the individual hypotheses along with the respective construct. 

6- The results debate and discuss the veracity of several indirect relationships however it is surprising that no indirect relations are claimed under hypotheses section. Why is that so? 

7- The results indicate a very strong correlation between SE and SR (0.57**) which is also reflective in table 3 where the discriminant validity measure fails to prove as the sq.root of AVE is 0.66 while the correlation value is 0.70 between these constructs. Authors should consider double-checking these two constructs on the basis of each item.  The removal of some highlighted correlated items would improve the results. These two constructs are the major focus of this study and hence should be dealt with more concern. I believe more statistical tests need to performed to ensure that this high correlation between two important variables will not affect the overall results and model fitness etc.

8-  The discussion section puts enough light on explaining the findings of the study but unfortunately I cannot see the research implication for theory and practice. A little effort on that part will make it more useful for the readers.

Author Response

Thank you very much for your review and constructive comments. We have made revisions according to your comments. The details are as follows.

RESPONSE to the 1st comment:

The subject in this study refers to those took part in classroom learning through the use of technology, we have clearly pointed this out in the article in line 32-33.

RESPONSE to the 2nd comment:

Thank you for raising this issue, we have carefully revised these problems. To be more specific, in the introduction section, we have indicated that most of previous studies were limited by the perspective of others, such as technology-centered perspective, and paid little attention to the initiative and responsibility of the students who are the protagonists of the learning process in online learning. Next, we introduce two additional variables of this paper based on the student’s perspective, which is also the basis for introduction of self-efficacy and self-regulation. The novelty of this paper is that it combines the variables extracted from the TAM,focusing on the adopted technonology and the two variables (self-regulation and self-efficacy) focusing on the students. As for the research contribution, we put it in the conclusion section and made corresponding modifications.

RESPONSE to the 3rd comment:

(1) We chose covariance-based SEM mainly for two reasons. First, previous studies (eg, Reinartz, Haenlein, & Henseler, 2009) have pointed out that covariance-based SEM should be the method of choice when the focus lies on confirming theoretically assumed relationships, and the sample size exceeds a certain threshold (250 observations). Second, previous many relevant studies have also adopted this method, eg, Lee & Park (2017), Wang & Zhao (2021).

(2) In this study, the sample size is 900. Because of the widespread of COVID-19, the selection of samples and the way of carrying out investigation are affected. We had to give out online questionnaire because offline face-to-face investigation is not allowed because of COVID-19, so it is not easy and convenient to get reliable samples in large scale.

Before we begin the investigation, we preliminarily determined the reference value of the sample size according to Gpower and traditional practices (for example, the number of samples should be at least 5 to 10 times the total number of survey scale items).  There are 25 items in our questionnaire, so the sample number should be at least 125 to 250. If we take the total number of college and university students in China into consideration, 125 to 250 participants will not be representative, thus we thought the sample should be larger than that. Then we drew experience from other similar studies. Finally, limited by time and cost, we got 900 questionnaires, among which 767 are valid. We’ve done our best to make samples representative in order to make the study result reflecting the fact.

In addition, we determined the sample based on the principles of convenience, randomness and purpose, we believe that students those from different provinces are more representative of the online learning situation of Chinese college students than groups with too high homogeneity from the same region, because these provinces in the study include educationally developed and underdeveloped regions, the eastern, central and western regions. Of course, there is no doubt that your views are very enlightening, however, determining quotas is a very troublesome issue, because it involves many factors, such as the population of each province, the level of economic development, the number of local universities and the scale of universities. The process of collecting a lot of this kind of information to calculate the quota is likely to make the article stagnant and unable to move forward. In addition, we have never found that the relevant research in China has adopted the quota method, so it is difficult for us to find relevant reference data, so we were not able to modify this content according to your suggestion.

RESPONSE to the 4th comment:

We agree with your opinion, which is also validated from the data itself, so we re-run the SEM with gender, ethnicity, and grade as control variables, with specific results in Figure 3, Table 4, and Table 5.

RESPONSE to the 5th comment:

We very much agree with your opinion and we have put the individual hypotheses along with the respective construct.

RESPONSE to the 6th comment:

As you said, the statement about indirect relationship appeared in the results and discussion but not in the hypothesis section, this is because the exploration of indirect relationship is not the focus of the study, we only present this part to illustrate the importance of some variables.

RESPONSE to the 7th comment:

We agree with you that high correlation and low discriminant validity between SE and SR may affect the overall results of the study, so we followed your suggestion and checked the items in both structures and removed one item, resulting in improved results in this section, including that the correlation between constructs is now less than the square root of the AVE. Of course, due to the deletion of an item, some other results have also changed to some extent.

RESPONSE to the 8th comment:

We have placed the research implication for theory and practice in the conclusion section and have made revisions in this version.

In addition, the manuscript has been checked by a colleague fluent in English writing.

Round 2

Reviewer 3 Report

The commented revision was checked and improved for publication. 

Author Response

Thank you. We checked the article again and improved it.

Reviewer 4 Report

The authors have improved the manuscript considerably but still some points need their attention.  I would recommend working on these below.

1-Co-variance based SEM methods are well-suited even in the presence of data normality issues and guard against all such factors even in the case of limited sample size. Covariance based methods are also recommended for research with closely related and correlated constructs as in this case. You may see for example https://doi.org/10.3390/su14095662 for reference.

2-The discussion on the sample size should clearly advocate for the points explained here. I believe the samples from different provinces might not be as diverse but still the student seniority (semester or year of study) and student personal experience with online tools would definitely have a say in this regard. If the study has not captured these variables, then it might be an opportunity for the future researchers to explore these uncovered variables in future.

3-The results should clearly mention the effects of these control variables which I believe are not accounted for hence leaving the reader to wonder about the influence of these control variables.

4- I disagree here because the presence of mediating relations clearly shows that some relations are more meaningful in the presence of mediation. The authors have emphasized too much on the direct relations which gives an impression that every variable has an influence on every other variable that makes little sense even if they so proven by the analysis. 

5-Since SE and SR are highly correlated hence, authors should check for multicollinearity issue for their parameters as well. I would also recommend for testing of Common method bias testing in this case.

Author Response

Thank you very much for your constructive comments. We have made revisions according to your comments.

RESPONSE to the 1st comment:

Thank you very much for providing this paper, which pointed out that “The choice of variance-based SEM technique was made based on such specific conditions, such as sample size, data normality, etc…we made use of Warp-PLS-7 for conducting the analysis, which is considered to be a robust and advanced tool under such conditions (some constructs reported non-normal data)”, which undoubtedly deepens our understanding of SEM selection criteria. Due to the importance of this article, we decided to refer to it in this article (line 373-376).

RESPONSE to the 2nd comment:

We have read your comments carefully and believe your comments are justified. Compared with other aspects (e.g., gender, ethnicity), the student seniority and student personal experience with online tools may have a substantial impact on the variables of this study. Unfortunately, we were unable to design and collect these information because we initially failed to attention the effect of them on online learning. Now, we cannot supplementally collect these data from the original subjects since the questionnaires have been finished. Given the opportunity, we will focus and explore these important variables and their effects in the future.

RESPONSE to the 3rd comment:

We agree with your suggestion, so we have added relevant content (including a paragraph and a table) about control variables to the results section, see line 359-370 for details.

RESPONSE to the 4th comment:

We have discussed the comment in depth, and finally we think that your suggestion is indeed reasonable. As you said, the significance of the study is reduced by only talking about direct effects. If indirect relationships are emphasized, the paper will be able to explore the significance of some variables and relationships more effectively. Therefore, we have revised the article according to your comments. The details are as follows:stating the results in the abstract (line 10-11), explicitly proposing the hypothesis about indirect effects in the literature review (line 221-239), and responding to the question about the validity of the hypothesis in the results section (line 404-407) and discussion (line 456-458).

RESPONSE to the 5th comment:

We have added tests for collinearity and common method bias to the article in line 339-354 based on your suggestion and with reference to the methods and criteria used in previous papers.

Round 3

Reviewer 4 Report

he authors have made good efforts to improve the quality of their manuscript. I still believe they can add more value by reducing the reducing the direct and indirect hypotheses. They should only focus on those relationships which are either

1. quantitatively more significant,

or

2. opens new avenues for researchers because of being untested before etc.

Or

3. supports the overall development framework of the study.

Author Response

Thank you for raising this issue. We very much agree with your opinion and do find that there are too many hypotheses in the original article; therefore, we decided to remove some hypotheses.

The variables in this manuscript are composed of two parts. Some variables (BI, PU and PEU) chosen from TAM forms the basis of the study. The exploration of these variables is very important, not only because it enables dialogue with previous articles but also because the epidemic is an important background of the times; it has become the most important influencing factor for students to choose online learning both in the past and currently. It may even continue to exert such influence in the future. Therefore, it is important to pay attention to and examine variables related to online learning. Finally, we decided to retain all assumptions about them. Some variables (SR and SE) come from the perspective of students, which is where this study tries to expand on previous studies. Although we think these selections are meaningful, the purpose should be to examine their impact on TAM variables, not the internal relationship between the two variables. Therefore, we decided to delete the hypotheses involving SR and SE at the same time, including H7, H14 (3), H14 (5), H14 (6), H15 (2), H15 (3), and H16.

After revising the article according to your comments, we found that reducing some hypotheses not only made the article more concise but also has the following advantages: (1) The article now has a larger proportion of significant hypotheses; (2) Partial variables (PU) play a more prominent role in the model.

See lines 9-11, 183, 195, 201, 225-233, 379-389, 390-391, 395-403, 411-412, 432-453, 454-456, and 460-461 for specific modifications.
